# Coping Strategies and Mental Disorders among the LGBT+ Community in Malaysia

**DOI:** 10.3390/healthcare10101885

**Published:** 2022-09-27

**Authors:** Johan Ariff Juhari, Jesjeet Singh Gill, Benedict Francis

**Affiliations:** 1Department of Psychological Medicine, University Malaya Medical Centre, Kuala Lumpur 59100, Malaysia; 2Department of Psychological Medicine, Faculty of Medicine, University Malaya, Kuala Lumpur 50603, Malaysia

**Keywords:** mental disorders, LGBT, Malaysia, coping

## Abstract

The LGBT+ community in Malaysia is vulnerable to mental disorders due to the pressures of being in a conservative heteronormative culture. This study aimed to study the association between coping strategies as well as the sociodemographic factors of LGBT+ individuals with mental disorders and quantify the occurrence of mental disorders among them. This study used a cross-sectional design. The self-rated Brief Coping Orientation to Problem Experienced Inventory (Brief COPE) and the Mini International Neuropsychiatric Interview (MINI) were used to this end. A total of 152 participants were recruited. Among the participants, 67.8% used mainly problem-focused coping strategies, 29.6% employed emotion-based coping, and 6.6% used avoidance coping. The prevalence of mental disorders in general and major depressive disorder was much higher than in the general Malaysian population (80.3% and 40.1%, respectively). The only sociodemographic factor that was significantly associated with mental disorders was bisexuality. Problem-focused coping is associated with fewer mental disorders, and emotion-based coping is associated with a higher prevalence of mental disorders. More studies need to be conducted to better understand and better manage the mental disorders of the Malaysian LGBT+ community.

## 1. Introduction

For the general population in Malaysia, mental disorders are becoming more prevalent. In comparison with the mental health problems of populations above the age of 16 in 1996 (10.7%), the National Health and Morbidity Survey in 2015 by the Ministry of Health of Malaysia recorded a nearly threefold increment up to 29.2% [1]. The 2019 National Health and Morbidity Survey by the same body recorded that the prevalence of depression in the general population was 2.3% [2].

LGBT+ is an umbrella term that encompasses spectrums of sexuality and genders, which includes but is not limited to homosexuals (lesbians and gays), bisexuals, and transgender individuals [3]. The term originally came from the initialism for lesbians, gays, bisexuals, and transgender individuals (LGBT) in the mid-to-late 1980s, but it is currently used to describe anybody who is non-heterosexual or non-cisgender instead of only the above groups [4]. To recognize this inclusion, there are other terms such as LGBT and questioning or queer individuals (LGBTQ), LGBT and intersex individuals (LGBTI), LGBT, questioning or queer, intersex, and asexual, agender, or aromantic (LGBTQIA), and other extensions to the initial abbreviation. For ease of reference, we have adapted the term LGBT+, where the ‘+’ represents those who are part of the community whose identity is not accurately captured or reflected by the term LGBT.

A 27-country survey performed by Ipsos in 2021 estimated that the global country average of individuals who identify their gender as transgender, nonbinary, nonconforming, gender-fluid, or in another way was about 1%. The same study found that 11% of the global country average identified as only, mostly, or equally attracted to the same sex. The survey also found that 1% of the population of Malaysia did not identify as cisgender, and 12% identified as only, mostly, or equally attracted to the same sex [5]. Other countries in the region such as the Philippines recorded 2.1% of the population as homosexual and 1.7% as bisexual [6], and Thailand was reported to have 5.8% of its population identifying as LGBT+ [7].

Malaysia is a conservative country where the majority of the public opinion of the LGBT+ community is not favorable. The Pew Research Centre reported that only 9% of the Malaysian population believe homosexuality should be accepted, in contrast to 86% [8] of the population being against the acceptance of homosexuality in society. A 2020 UCLA School of Law study [9] found that 41.1% of the Malaysian public find that society has gone too far in allowing transgender people to live and dress as their gender identity, 59.8% worry about exposing their children to transgender people. and 54.3% believe that transgender people are violating the traditions of their culture.

Beyond the societal stance on the LGBT+ community, the Malaysian legal system has laws encoded in both the penal code [10] and the state-level Syariah laws [11,12], where the former carries a maximum penalty of 20 years imprisonment with liability to fines and whipping [13]. Transgender individuals are also liable to be prosecuted under both civil laws as ‘public indecency’ [14] and state-level Syariah laws as ‘male person posing as a woman’ or ‘female person posing as a man’ [15]. 

The stigmas that Malaysian LGBT+ individuals face from society, the law, and within themselves are often caused by societal and religious values that are incompatible with non-heterosexual or non-cisgendered ideas. This then would lead to institutionalized prejudice, social stress, social exclusion, discriminatory hatred and violence, and internalized guilt and shame [16].

Multiple international studies have shown that people who self-identify as sexual minorities are at higher risk of developing mental disorders, suicidal ideation, and deliberate self-harm when compared with their heterosexual counterparts [16]. Lesbians, gays, and bisexual individuals are at least 1.5 times more likely to develop depression, anxiety, and substance dependence and twice as likely to have had suicidal attempts [16,17]. Among the transgender population, the likelihood to experience serious psychological distress is almost 8 times higher compared with the general population (39% vs. 5%), and the prevalence of lifetime suicide attempts is almost 10 times more (40% vs. 4.6%) [18]. In seeking healthcare, the LGBT+ population are more likely to report being dissatisfied with primary care and to have a negative experience when seeking healthcare [18,19].

Studies in Malaysia focusing on sexual minorities often focused on sexually transmitted diseases (41%), men who have sex with men (39%), and trans women (30%) [20]. However, there is no published study on the prevalence of mental disorders among the general Malaysian LGBT+ population.

This study aimed to obtain a more inclusive picture of the mental health status of the LGBT+ community in Malaysia. As the LGBT+ community in Malaysia is a vulnerable group that faces stigmas from themselves, family, society, and institutions, it is imperative that we have a better understanding of the struggles and coping mechanisms that they employ. Knowing the occurrence of mental disorders in the LGBT+ community and their coping strategies for those issues could aid mental health professionals to tailor specific risk stratification and intervention for this vulnerable population. As such, it is the objective of this study to study the association between coping strategies as well as the sociodemographic factors of LGBT+ individuals with mental disorders and quantify the occurrence of mental disorders among them. This study also seeks to stratify the rates of occurrence of specific mental disorders. For the purpose of this study, a mental disorder is defined as the conditions that fulfil the diagnostic criteria of mental disorders in the Diagnostic and Statistical Manual of Mental Disorders (DSM) that are also included in the Mini International Neuropsychiatric Interview (MINI).

## 2. Materials and Methods

This study was conducted cross-sectionally and based on the community. An online recruitment advertisement was created and disseminated through engagement with local LGBT+ non-governmental organizations such as PLUHO, Jejaka, Seksualiti Merdeka, and others, as well as through social media (i.e., Twitter, Facebook, Telegram groups, etc.). Due to the sensitive nature of the study and potential samples, this study had to resort to snowball sampling, a non-probability sampling method. Those who were interested would voluntarily contact the investigators via the advertisements, and a thorough explanation regarding the study would be given. The individuals who consented to participating in the study were then assessed based on the inclusion and exclusion criteria. In total, 155 people were recruited, and from that group, 3 people were excluded from the study due to the exclusion criteria.

### 2.1. Inclusion Criteria

The subject is a Malaysian citizen;The subject is aged 18 years and above;The subject identifies as either a sexual minority or a transgendered individual;The subject is capable of understanding and answering questions in English or Malay;The subject gives consent to participate in this study.

### 2.2. Exclusion Criteria

The subject does not give or withdraws their consent to participate in the study;The subject is not able to understand and answer questions in English or Malay.

A preconstructed questionnaire that was designed for the research was used to gather sociodemographic data. Other than the sociodemographic data, also included were Likert scales to score families’ knowledge and support of the participants’ sexual orientations or gender identities, friends’ knowledge and support of the participants’ sexual orientations or gender identities, and colleagues’ knowledge and support of the participants’ sexual orientations or gender identities.

The coping skills of the subjects were measured using the Brief Coping Orientation to Problem Experienced Inventory (Brief COPE). Upon completion of these two self-rated questionnaires, the subjects were then interviewed using a structured diagnostic interview: the Mini International Neuropsychiatric Interview (MINI). This study was approved by the Medical Ethics Committee of University Malaya (MREC 20201223-9616).

### 2.3. Measurement Tools

#### 2.3.1. Brief COPE

The Brief COPE is a self-rated questionnaire designed to measure different ways people emotionally respond to a serious circumstance. There are 28 items concerning 3 primary coping styles: problem-focused coping, emotion-focused coping, and avoidant coping. These coping styles could be further subdivided into 14 different coping skills: denial, substance abuse, self-distraction, behavioral disengagement, emotional support, humor, venting, self-blame, acceptance, religion, active coping, use of instrumental support, planning, and positive reframing [21]. It has been adapted and validated to the Malaysian population, with the total Cronbach’s value for the internal consistency being 0.83 [22].

#### 2.3.2. MINI

The MINI is a diagnostic structured interview that was designed to provide a diagnosis based on the DSM. It contains 17 different modules covering major mental illnesses, including major depressive disorder, bipolar disorders, anxiety disorders, and eating disorders. This clinician-rated interview can be administered in a shorter time compared with other similar instruments, such as the CIDI and SCID-P. [23] The MINI has also been adapted and validated to the Malaysian population, with Fleiss’ kappa showing values ranging between 0.67 and 0.85 (satisfactory to excellent) [24].

### 2.4. Statistical Analyses

Data analysis was executed with Statistical Package for the Social Sciences (SPSS) version 24 (IBM Corp., Armonk, NY, USA). This study will give results for the sociodemographic characteristics of the participants in the LGBT+ community, which are presented as counts with percentages for categorical variables. Chi-square analysis was employed to identify the association of sociodemographic factors with mental disorders among the LGBT+ community. The simple logistics regression and proportional hazard model were analyzed to determine which independent variable had an intensifying effect on mental disorders when analyzed as a set and compute their statistical significance within a confidence interval (CI) of 95%. The result of *p* values of less than 0.05 from simple logistics regression was then tested further via multiple logistics regression. Spearman rank correlation analysis was used to examine the association between the scoping scale (problem-focused, emotion-focused, and avoidant-focused) and the mental disorders of the participants among the LGBT+ community. In this study, two-tailed comparative analysis was used, and the statistical significance was defined as *p* < 0.05.

## 3. Results

A total of 155 individuals were recruited for this study. One participant was excluded as they did not meet the inclusion criteria, and two had retracted their consent. As such, a total of 152 participants were analyzed in this study.

### 3.1. Demographic Data

Table 1 demonstrates the sociodemographic information of the participants in the LGBT+ community in this study. The majority of the participants who were involved in this study were between 25 and 35 years old (64.5%), followed by those under 25 years old (27.0%) and more than 35 years old (8.6%). More males (59.9%) than females (20.4%) joined the study, and most of the participants had a homosexual orientation (59.9%). There were four participants that identified as heterosexual. Of these, three of them identified as trans men, and one identified as other. About 39.5% and 36.2% of the participants in this study were from Chinese and Malay populations, respectively. Most of them resided in the urban area (69.1%) and were from states in Semenanjung Malaysia (67.1%), which excludes the Federal Territories (FTs), or Wilayah Persekutuan.

Most of the participants came from a middle-income background (42.8%), followed by low-income (38.2%) and then high-income (15.1%) backgrounds. This division of household income is in accordance with the income classification by the Department of Statistics Malaysia [25]. The highest level of education that most of the participants had was a tertiary education background (91.4%), followed by a secondary education background (8.6%). About 64.5% of them were working, while 23.7% were still studying and the others were unemployed.

The Brief COPE is a 28-item self-reporting questionnaire designed to measure the effectiveness and ineffectiveness of ways to cope with stressful life events, which comprise problem-focused coping, emotion-focused coping, and avoidant coping. From this study, about 67.8%, which represented 103 participants, practiced problem-focused coping. The results also show that 45 (29.6%) participants used emotion-focused coping, while 10 participants (6.6%) practiced avoidant coping.

### 3.2. Prevalence of Mental Disorders

Table 2 describes the prevalence of mental disorders as found in the MINI among the participants of the study. Major depressive disorder had the highest prevalence among the listed mental disorders at 40.1%, followed by agoraphobia at 22.4% and suicidal behavior disorder at 21.1%.

Of the 152 participants, 122 (80.3%) of them were found to have been diagnosed with at least 1 mental disorder as per the MINI.

### 3.3. Chi-Square Analysis for the Risk Factors of Mental Disorders

Chi-square analysis was performed to identify the association of sociodemographic data and the primary coping styles in relation to the prevalence of mental disorders among the participants as presented in Table 3. From the analysis, the results show that the area of living (*p* = 0.042), family knowledge (*p* = 0.016), friends’ knowledge (*p* = 0.040), friends’ support (*p* = 0.045), problem-focused coping (*p* = 0.013), and emotion-focused coping (0.029) were significant in relation to the prevalence of mental disorders among the participants. Since the *p* value for all these risk factors was less than our chosen significance level (α = 0.05), we concluded that there is enough evidence to suggest an association between those risk factors and mental disorders among the LGBT+ community.

Aside from that, the others risk factors (i.e., sexual orientation (*p* = 0.054), age group (*p* = 0.0624), gender (*p* = 0.317), ethnicity (*p* = 0.209), religion (*p* = 0.129), states (*p*= 0.350), education (*p* = 0.680), occupation (*p*= 0.254), colleagues’ knowledge (*p*= 0.249), colleagues’ support (*p*= 0.352), income (*p* = 0.552), family support (0.943), and avoidant-coping (*p* = 0.103)) gave results over our chosen significance level (α = 0.05), and thus these factors were not statistically significant. As such, it can be concluded that there is enough evidence to suggest that there is no association between those risk factors and the prevalence of mental disorders among the LGBT+ community.

### 3.4. Simple and Multiple Logistics Regression for the Risk Factors of Mental Disorders

The analysis of logistic regression was performed to ascertain the effects of sociodemographic and primary coping styles on the likelihood of having mental disorders among the participants and is represented in Table 4. Based on the results, identifying as bisexual was statistically significant (*p* = 0.043) in having a mental disorder. It can be concluded that the bisexual subgroup was 3.721 times more likely to exhibit mental disorders compared with the other subgroups. Multiple logistics regression analysis was performed with the result of an adjusted OR of 1.097 (*p* = 0.048).

The results also show that the lack of friends’ knowledge about the participant’s gender identity or sexual orientation was statistically significant (*p* = 0.013). The participants whose friends did not know their gender identity or sexual orientation were 0.052 times as likely to have mental disorders compared with those whose friends did know about their gender identity or sexual orientation. Multiple logistics regression analysis was performed with the result of an adjusted OR of 37.475 (*p* = 0.011).

Based on the primary coping styles, the results show that those who adopted emotion-focused coping (*p* = 0.037) were statistically significant in having mental disorders, as well as those who did not adopt problem-focused coping (*p* = 0.019). Those with emotion-focused coping were 3.290 times more likely to exhibit mental disorders. Similarly, those who did not adopt problem-focused coping were 3.799 times more likely to exhibit mental disorders. Multiple logistics regression analysis did not yield statistically significant results.

### 3.5. Spearman’s Rank Correlation Analysis for Correlation of the Primary Coping Styles with the Mental Disorders of the Participants among the LGBT+ Community

Spearman’s rank correlation analysis was used to assess the monotonic relationship between the primary coping styles with respect to the mental disorders of the participants among the LGBT+ community as can be seen in Table 5. Correlation analysis demonstrated a significantly negative correlation for problem-focused coping (rs = −0.201, *p* = 0.013) and significantly positive correlation for emotion-focused coping (rs = 0.177, *p* = 0.029) with respect to the prevalence of mental disorders among the participants.

## 4. Discussion

The LGBT+ community has been recognized worldwide to be more vulnerable to mental disorders compared with the general population [16,17,18,19,20]. Despite that, there is a dearth of critical data on this subject in Malaysia and the Southeast Asia region. Most studies regarding mental health when it comes to the LGBT+ population tend to come from the high-income countries of North America and Europe [26]. As such, these data might not apply to an economically developing [27] and conservative nation like Malaysia. Local Malaysian studies tend to focus more on sexually transmitted diseases and subjects who are assigned male at birth [20]. Given the added societal [8,9] and legal [10,11,12] pressure in the Malaysian conservative society [8], the prevalence of mental disorders in this vulnerable community would be even more pronounced in comparison with the general Malaysian population.

Our findings suggest that Malaysian LGBT+ individuals with problem-focused coping are significantly less likely to have mental disorders (with a rank correlation of rs = −0.201, *p* = 0.013), while those who employ emotion-focused coping are more likely to have mental disorders (rs = 0.177, *p* = 0.029). Problem-focused coping puts emphasis on efforts to modify the immediate issues and usually involves finding routes to solve the problem, comparing the pros and cons of different choices, and taking steps to solve the problem [28]. It consists of active coping, the use of informational support, positive reframing, and planning. [21] Problem-focused coping has been shown to significantly and positively impact the quality of life of patients with fibromyalgia [29].

Emotion-focused coping, on the other hand, aims to handle the emotional distress that is associated with a particular situation [28]. It attempts to do so through emotional support, venting, humor, acceptance, religion, and self-blame [21]. The effectiveness of emotion-focused coping depends highly on the specific strategy employed, and as such, the stress and coping literature predominantly views emotion-focused coping as maladaptive [30].

Baker et al. showed that while emotion-focused coping could increase insight and causal thinking, they also found that engaging in this type of coping was less adaptive when the participants had interpersonal stressors. They hypothesized that over-employment of emotion-focused coping may overwhelm and cause conflict with one’s social support system, which leads to a consequent decrease in positive effects [30]. This could explain our findings, as mental disorders often have a two-way association with one’s interpersonal relationships. Those that employ emotion-focused excessively would cause conflict with their support system, which in turn would affect their mental health.

It is also of note that our findings indicate that the prevalence of mental disorders among the Malaysian adult LGBT+ population is more than double that of the general population (80.3% vs. 29.2%) [1], which is in line with international findings [16,17,18]. In any society, and even more so in a conservative society such as Malaysia, LGBT+ individuals face stigmas from society and within themselves. This is often caused by heteronormative societal and religious values that are incompatible with non-heterosexual or non-cisgendered ideas. LGBT+ individuals are made more vulnerable to mental disorders as they face institutionalized prejudice, social stress, social exclusion, discriminatory hate and violence, and internalized guilt and shame [31,32].

Among the specific mental disorders, the one with the highest prevalence is major depressive disorder, which is at least 3 times (40.1% vs. 12%) [33] and as high as 20 times (40.1% vs. 2.3%) [2] more prevalent than in the Malaysian general population. This is comparable to a Thai study with similar populations, where the prevalence of depression was found to be 40.3% [34], and a multinational Southeast Asian study, which found depression to be in 23.5% of its sample [35]. Although there is no recent source for the prevalence of suicidal behavior disorder or suicide attempts for the general Malaysian population, our study found that 21.1% of our participants fulfilled the criteria of suicidal behavior disorder, compared with the Thai data of 13.1% [36] and the multinational Southeast Asian data of 35.3% [35].

Surprisingly, the risk factors that are associated with depression in the Malaysian population, such as living in rural areas, unemployment, and a lower household income group [2], are not significantly associated with the prevalence of major depressive disorder in the Malaysian LGBT+ community (results not shown here).

The only sociodemographic factor that seemed to be statistically significant in association with the presence of mental disorders was bisexuality. Multiple studies have shown that people who identify with being bisexual have poorer mental health compared with other sexual orientation groups [37,38,39,40,41]. People who identified as bisexual and those who reported attraction to more than one gender were found to have higher rates of mood disorders, according to an American study [38]. Jorm et al. suggested that bisexuality itself might be a risk factor in having mental disorders. Their findings suggested that poor mental health in homosexual participants could be accounted for by sociodemographic risk factors and early life psychosocial experiences, whereas these factors are not statistically significant in bisexual people [40].

## 5. Limitations

There are several limitations that need to be acknowledged in this study. The cross-sectional nature of this study prevented us from assessing the causality of the factors; instead, we could only study their association. We used convenient snowball sampling to collect our samples from this vulnerable population, as there is a very real fear of danger and legal ramifications in identifying as part of the LGBT+ community in Malaysia. The sampling techniques could be improved to reach a more accurate prevalence of data. Advertisements were distributed through online channels, such as social media, WhatsApp, and Telegram messages, as well as through non-governmental organizations, such as PLUHO, Jejaka, and others. As such, our study had difficulty permeating to non-urban areas and less developed states. Having more participants from these areas would have given a clearer picture of the results.

## 6. Conclusions

To the best of our knowledge, this is the first study in Malaysia that assessed the coping styles and mental disorders among the general Malaysian LGBT+ community. The high prevalence of mental disorders in general, and specifically major depressive disorder, is especially worrying. Despite this, the usual sociodemographic factors that are commonly associated with poor mental health and depression did not seem to be significant statistically. Societal and legal pressures and the lack of basic human rights protection for the LGBT+ community could potentially be the driving force that affects the mental health of the community negatively. A lot more studies need to be conducted to better understand and manage the mental disorders of the LGBT+ community in Malaysia.

## Figures and Tables

**Table 1 healthcare-10-01885-t001:** Sociodemographic data and coping skills among the LGBT+ community in Malaysia.

	N = 152	Percent = 100%
**Age Group**		
<25 years old	41	27.0
25–35 years old	98	64.5
>35 years old	13	8.6
**Gender**		
Male	91	59.9
Female	31	20.4
Trans man	7	4.6
Trans woman	5	3.3
Others	18	11.8
**Sexual Orientation**		
Asexual	4	2.6
Bisexual	36	23.7
Heterosexual	4	2.6
Homosexual (lesbian or gay)	91	59.9
Others	17	11.2
**Ethnicity**		
Malay	55	36.2
Chinese	60	39.5
Indian	20	13.2
Bumiputra Sabah and Sarawak	14	9.2
Other	3	2.0
**Religion**		
Islam	47	30.9
Atheism	19	12.5
Buddhism	25	16.4
Christianity	31	20.4
Hinduism	9	5.9
Others	21	13.8
**Area of Living**		
Urban area	105	69.1
Suburban area	41	27.0
Rural area	6	3.9
**States**		
Semenanjung states (excluding FT)	102	67.1
Sabah and Sarawak	6	3.9
Wilayah Persekutuan	38	25.0
Outside of Malaysia	6	3.9
**Education**		
Secondary	13	8.6
Tertiary	139	91.4
Occupation		
Working	98	64.5
Studying	36	23.7
Unemployed	18	11.8
**Colleagues’ Knowledge**		
No	16	10.5
Yes	30	19.7
Some or maybe	52	34.2
N/A	54	35.5
**Colleagues’ Support**		
High	56	36.8
Moderate	33	21.7
Low	9	5.9
N/A	54	35.5
**Income ***		
Unwilling or unable to disclose	6	3.9
B40	58	38.2
M40	65	42.8
T20	23	15.1
**Family Knowledge**		
No	47	30.9
Yes	46	30.3
Some or maybe	59	38.8
**Family Support**		
High	32	21.1
Moderate	57	37.5
Low	63	41.4
**Friends’ Knowledge**		
No	4	2.6
Yes	89	58.6
Some or maybe	59	38.8
**Friends’ Support**		
High	117	77.0
Moderate	30	19.7
Low	5	3.3
**Problem-Focused Coping**		
No	49	32.2
Yes	103	67.8
**Emotion-Focused Coping**		
No	107	70.4
Yes	45	29.6
**Avoidant Coping**		
No	142	93.4
Yes	10	6.6

* Based on the Department of Statistics Malaysia, the financial classification of monthly household income was as such: B40 = MYR 4849 or less, M40 = between MYR 4850 and 10,959, and T20 = MYR 10,960 or more.

**Table 2 healthcare-10-01885-t002:** Prevalence of specific mental disorders among the LGBT+ community in Malaysia.

	**N = 152**	**Percent = 100%**
**Major Depressive Disorder, All**		
No	91	59.9
Yes	61	40.1
**Suicidal Behavior Disorder, All**		
No	120	78.9
Yes	32	21.1
**Bipolar Disorder, All**		
No	128	84.2
Yes	24	15.8
**Panic Disorders, All**		
No	140	92.1
Yes	12	7.9
**Agoraphobia, Current**		
No	118	77.6
Yes	34	22.4
**Social Anxiety Disorder, Current**		
No	138	90.8
Yes	14	9.2
**Obsessive Compulsive Disorder, Current**		
No	142	93.4
Yes	10	6.6
**Post-Traumatic Stress Disorder, Current**		
No	143	94.1
Yes	9	5.9
**Alcohol Use Disorder, Past 12 Months**		
No	132	86.8
Yes	20	13.2
**Substance Use Disorder, Past 12 Months**		
No	127	83.6
Yes	25	16.4
**Psychotic Disorder, All**		
No	150	98.7
Yes	2	1.3
**Bulimia Nervosa, Current**		
No	149	98.0
Yes	3	2.0
**Generalized Anxiety Disorder, Current**		
No	130	85.5
Yes	22	14.5
**Organic Causes**		
No	151	99.3
Yes	1	0.07
**Antisocial Personality Disorder**		
No	151	99.3
Yes	1	0.07
**Presence of Mental Disorders**		
No	30	19.7
Yes	122	80.3

**Table 3 healthcare-10-01885-t003:** Association of sociodemographic and primary coping styles among the LGBT+ community in Malaysia.

Demographic Variables	Mental Health Disorder		Test Statistics	*p* Value
	No	Yes		
**Age Group**				
<25 years old	6 (14.6)	35 (85.4)	0.942	0.624
25–35 years old	21 (21.4)	77 (78.6)		
>35 years old	3 (23.1)	10 (76.9)		
**Gender**				
Male	23 (25.3)	68 (74.7)	4.719	0.317
Female	3 (9.7)	28 (90.3)		
Trans man	1 (14.3)	6 (85.7)		
Trans woman	1 (20.0)	4 (80.0)		
Others	2 (11.1)	16 (88.9)		
**Sexual Orientation**				
Asexual	1 (25.0)	3 (75.0)	9.159	0.054
Bisexual	3 (8.3)	33 (91.7)		
Heterosexual	2 (50.0)	2 (50.0)		
Homosexual (lesbian or gay)	23 (25.3)	68 (74.7)		
Others	1 (5.9)	16 (94.1)		
**Ethnics**				
Malay	8 (14.5)	47 (85.5)	5.875	0.209
Chinese	17 (28.3)	43 (71.7)		
Indian	3 (15.0)	17 (85.0)		
Bumiputra Sabah and Sarawak	1 (7.1)	13 (92.9)		
Other		3 (100.0)		
**Religion**				
Islam	6 (12.8)	41 (87.2)	8.540	0.129
Atheism	3 (15.8)	16 (84.2)		
Buddhism	10 (40.0)	15 (60.0)		
Christianity	6 (19.4)	25 (80.6)		
Hinduism	1 (11.1)	8 (88.9)		
Others	4 (19.0)	17 (81.0)		
**Area of Living**				
Urban area	23 (21.9)	82 (78.1)	6.359	0.042 ^1^
Suburban area	4 (9.8)	37 (90.2)		
Rural area	3 (50.0)	3 (50.0)		
**States**				
Semenanjung states (excluding FT)	23 (22.5)	79 (77.5)	3.280	0.350
Sabah and Sarawak	2 (33.3)	4 (66.7)		
Wilayah Persekutuan	4 (10.5)	34 (89.5)		
Outside of Malaysia	1 (16.7)	5 (83.3)		
**Education**				
Secondary	2 (15.4)	11 (84.6)	0.170	0.680
Tertiary	28 (20.1)	11 (79.9)		
**Occupation**				
Working	22 (22.4)	76 (77.6)	2.742	0.254
Studying	7 (19.4)	29 (80.6)		
Unemployed	1 (5.6)	17 (94.4)		
**Colleagues’ Knowledge**				
No	3 (18.8)	13 (81.2)	2.780	0.249
Yes	4 (13.3)	26 (86.7)		
Some or maybe	15 (28.8)	37 (71.2)		
**Colleagues’ Support**				
High	11 (19.6)	45 (80.4)	2.087	0.352
Moderate	10 (30.3)	23 (69.7)		
Low	1 (11.1)	8 (88.9)		
**Income**				
B40	10 (17.2)	48 (82.8)	1.187	0.552
M40	16 (24.6)	49 (75.4)		
T20	4 (17.4)	19 (82.6)		
**Family Knowledge**				
No	14 (29.8)	33 (70.2)	8.227	0.016 ^1^
Yes	11 (23.9)	35 (76.1)		
Some or Maybe	5 (8.5)	54 (91.5)		
**Family Support**				
High	7 (21.9)	25 (78.1)	0.118	0.943
Moderate	11 (19.3)	46 (80.7)		
Low	12 (19.0)	51 (81.0)		
**Friends’ Knowledge**				
No	3 (75.0)	1 (25.0)	11.113	0.040 ^1^
Yes	12 (13.5)	77 (86.5)		
Some or maybe	15 (25.4)	44 (74.6)		
**Friends’ Support**				
High	18 (15.4)	99 (84.6)	6.196	0.045 ^1^
Moderate	10 (33.3)	20 (66.7)		
Low	2 (40.0)	3 (60.0)		
**Problem-Focused Coping**				
No	4 (8.2)	45 (91.8)	6.114	0.013 ^1^
Yes	26 (25.2)	77 (74.8)		
**Emotion-Focused Coping**				
No	26 (24.3)	81 (75.7)	4.749	0.029 ^1^
Yes	4 (8.9)	41 (91.1)		
**Avoidant Coping**				
No	30 (21.1)	112 (78.9)	2.632	0.103
Yes		10 (100.0)		

* Based on the Department of Statistics Malaysia, the financial classification of monthly household income is as such: B40 = MYR 4849 or less, M40 = between MYR 4850 and 10,959, T20 = MYR 10,960 or more. ^1^
*p* < 0.05.

**Table 4 healthcare-10-01885-t004:** Association of sociodemographic and primary coping styles with mental disorders among the LGBT+ community in Malaysia.

Factors	Crude OR (95% CI)	*p* Value	Adjusted OR (95% CI)	*p* Value
**Age Group**				
<25 years old (R)				
25–35 years old	1.591 (0.590, 4.287)	0.359		
>35 years old	0.909 (0.229, 3.604)	0.892		
**Gender**				
Male (R)				
Female	3.157 (0.877, 11.366)	0.079		
Trans man	2.0929 (0.232, 17.759)	0.523		
Trans woman	1.353 (0.144, 12.731)	0.792		
Others	2.706 (0.578, 12.674)	0.206		
**Sexual Orientation**				
Homosexual (Lesbian or gay) (R)				
Asexual	1.015 (0.101, 10.243)	0.990	0.142 (0.006, 3.599)	0.237
Bisexual	3.721 (1.042, 13.288)	0.043 ^1^	1.097 (0.089, 13.560)	0.048 ^1^
Heterosexual	0.338 (0.045, 2.540)	0.292	0.076 (0.004, 1.356)	0.080
Others	5.412 (0.680. 43.096)	0.111	0.150 (0.018, 1.251)	0.080
**Ethnics**				
Chinese (R)				
Malay	0.431 (0.169, 1.098)	0.078		
Indian	0.681 (0.181, 2.568)	0.570		
Bumiputra Sabah and Sarawak	2.213 (0.253, 19.335)			
Other				
**Religion**				
Islam (R)				
Atheism	0.780 (0.174, 3.503)	0.746		
Buddhism	0.220 (0.068, 0.709)	0.011		
Christianity	0.610 (0.177, 2.099)	0.433		
Hinduism	1.171 (0.124, 11.091)	0.891		
Others	0.622 (0.156, 2.486)	0.502		
**Area of Living**				
Urban area (R)				
Suburban area	2.595 (0.838, 8.036)	0.098		
Rural area	0.280 (0.053, 1.484)	0.135		
**States**				
Semenanjung states (excluding FT) (R)				
Sabah and Sarawak	0.582 (0.100, 3.384)	0.547		
Wilayah Persekutuan	2.475 (0.795, 7.702)	0.118		
Outside of Malaysia	1.456 (0.162, 13.094)	0.738		
**Education**				
Tertiary (R)				
Secondary	1.387 (0.291, 6.620)	0.681		
**Occupation**				
Working (R)				
Studying	1.199 (0.463, 3.107)	0.708		
Unemployed	4.921 (0.620, 39.071)	0.132		
**Colleagues’ Knowledge**				
No (R)				
Yes	1.757 (0.437, 7.063)	0.427		
Some or maybe	2.635 (0.785, 8.851)	0.117		
**Colleagues’ Support**				
High (R)				
Moderate	0.562 (0.208, 1.517)	0.256		
Low	1.956 (0.221, 17.315)	0.547		
**Income**				
B40 (R)				
M40	0.638 (0.263, 1.545)	0.638		
T20	0.990 (0.276, 3.543)	0.990		
**Family Knowledge**				
Yes (R)				
No	0.741 (0.295, 1.862)	0.741		
Some	3.394 (1.086, 10.608)	3.394		
**Family Support**				
High (R)				
Moderate	0.854 (0.294, 2.479)	0.772		
Low	1.016 (0.409, 2.525)	0.972		
**Friends’ Knowledge**				
Yes (R)				
No	0.052 (0.005, 0.541)	0.013 ^1^	34.475 (2.214, 536.886)	0.011 ^1^
Some or maybe	0.457 (0.196, 1.064)	0.069	12.388 (0.821, 186.812)	0.069
**Friends’ Support**				
High (R)				
Moderate	0.364 (0.146, 0.904)	0.029		
Low	0.273 (0.043, 1.749)	0.171		
**Problem-Focused Coping**				
Yes (R)				
No	3.799 (1.246, 11.585)	0.019^1^	0.221 (0.030, 1.633)	0.139
**Emotion-Focused Coping**				
No (R)				
Yes	3.290 (1.076, 10.060)	0.037 ^1^	0.867 (0.112, 6.697)	0.892
**Avoidant Coping**				
No (R)				
Yes				

* Based on the Department of Statistics Malaysia, the financial classification of monthly household income is as such: B40 = MYR 4849 or less, M40 = between MYR4850 and 10,959, and T20 = MYR 10,960 or more. ^1^
*p* < 0.05.

**Table 5 healthcare-10-01885-t005:** Spearman’s rank correlation between different coping styles and mental disorders among the LGBT+ community in Malaysia.

Problem-Focused Coping and Mental Disorders				
			Problem-Focused Coping	Mental Disorders
Spearman’s rho	Problem-Focused Coping	Correlation coefficient	1.000	−0.201
		Sig. (2-tailed)		0.013 **
		N	152	152
	Mental Disorders	Correlation coefficient	−0.201	1.000
		Sig. (2-tailed)	0.013 **	
		N	152	152
**Emotion-Focused Coping with Mental Disorders**				
			**Emotion-Focused Coping**	**Mental Disorders**
Spearman’s rho	Emotion-Focused Coping	Correlation coefficient	1.000	0.177
		Sig. (2-tailed)		0.029 **
		N	152	152
	Mental Disorders	Correlation coefficient	0.177	1.000
		Sig. (2-tailed)	0.029 **	
		N	152	152
**Avoidant Coping with Mental Disorders**				
			**Avoidant Coping**	**Mental Disorders**
Spearman’s rho	Avoidant Coping	Correlation coefficient	1.000	0.132
		Sig. (2-tailed)		0.106
		N	152	152
	Mental Disorders	Correlation coefficient	0.132	1.000
		Sig. (2-tailed)	0.106	
		N	152	152

** Correlation is significant at the 0.05 level (2-tailed).

## Data Availability

The data presented in this study are available upon request from the corresponding author. The data are not publicly available due to the sensitive nature of the population.

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
