# Peer review of "Coping Strategies and Mental Disorders among the LGBT+ Community in Malaysia"

_healthcare, 2022, doi:10.3390/healthcare10101885_

Round 1

Reviewer 1 Report

Dear Editor Dr. Neacsu Cristian, and dear authors, thank you for giving me the opportunity to review this article entitled "Prevalence of Mental Health Issues Among the LGBT+ Community in Malaysia". Overall, it seems to me that the article is minimally presentable, and could be improved for publication. However, it needs some general modifications, I present below a set of concerns and criticism that, I hope, will be useful for the authors in the process of editing the article:

Overall, I have three major concerns with this article. The first is that it is explicitly wrong for the authors to speak of the "prevalence" of any construct given that the sampling process was carried out using a convenience and snowball method. The title does not reflect the content of the article. This implies structural changes in the body of the study, and particularly in the title and the way in which the objectives are written. We cannot forget that this is mainly a descriptive study.

Second, on the basis of what model of psychopathology do you define "mental health issues"? It is a very broad term and, being the basis of the article, it has to be better substantiated in the introduction and throughout the article. I noticed that they used an instrument based on the dsm, and this already gives us hints about the classifying psychopathological approach that the authors opted for, but they need to be clearer. This is one of the aspects that cannot fail to explore.

Third, a methodological issue. Several instruments lack essential information. Were they adapted for the population in Malaysia? What are the psychometric indicators - mainly of reliability? Can you give some examples of items? What is the response scale to the items?

There are other small issues that also need to be reviewed, for example:

Table 3, in my opinion, should be eliminated. It does not provide significant information since the instruments used in the studies in the table are different from those used in this study. Furthermore, the criteria that led to the selection of these studies are not clear (Why these and not others? What are the selection criteria?). Also, the article has a lot of tabulated data, and this table causes even more noise to the reader.

Also, if the study refers to the LGBT population, why did it include 4 people who self-identify as heterosexual?

Since I suggest some significant and structural changes to the manuscript, I will gladly receive the authors' revisions and, then, with the text already reformulated, I will give some more minor suggestions to edit the article in order to the publication.

Reviewer 2 Report

Nicely constructed article. Authors can highlight the incidence of LBBTQ+ community in terms of total population and mental condition. Authors have not mentioned how dis they identified the individuals belonging to LGBTQ+ community. Whether they voluntarily came forward or how?

Round 2

Reviewer 1 Report

Dear authors, thank you for your work and the care with which you addressed the reflections I left you. With the changes made, it seems to me that the paper is in a position to be published. Good work.

Author Response

Thank you for your kind words.

Reviewer 2 Report

If the data in your country is not available, you can try and gather information in other country. At least it will give an idea to the readers and also helps to the status by comparing data of your country.

Author Response

Thank you for giving your valuable input in your review of this article. We shall try to address all the issues that you have suggested.

Point 1: If the data in your country is not available, you can try and gather information in other country. At least it will give an idea to the readers and also helps to the status by comparing data of your country.

Response 1: Your suggestion has been really useful, and we have taken it to improve our article. We have added the demographics data from international sources (lines 44 – 51). For the prevalence of mental health among the LGBT+ community, we have quoted multiple international sources in lines (71 – 80), as well as from the region (lines 314 – 322)

Thank you again for helping us improve the quality of our article. Your input has been invaluable to us.